# Development of a long noncoding RNA-based machine learning model to predict COVID-19 in-hospital mortality

Yvan Devaux [1] ✉, Lu Zhang[2], Andrew I. Lumley [1],
Kanita Karaduzovic-Hadziabdic[3], Vincent Mooser [4], Simon Rousseau [5],
Muhammad Shoaib [6], Venkata Satagopam [6], Muhamed Adilovic [3],
Prashant Kumar Srivastava[7], Costanza Emanueli [7], Fabio Martelli [8],
Simona Greco[8], Lina Badimon [9], Teresa Padro [9], Mitja Lustrek [10],
Markus Scholz [11], Maciej Rosolowski[11], Marko Jordan[10], Timo Brandenburger[12],
Bettina Benczik [13], Bence Agg [13], Peter Ferdinandy[13],
Jörg Janne Vehreschild [14,15,16,17], Bettina Lorenz-Depiereux[18], Marcus Dörr[19],
Oliver Witzke[20], Gabriel Sanchez[21], Seval Kul[21], Andy H. Baker [22,23],
Guy Fagherazzi[24], Markus Ollert [25,26], Ryan Wereski[27], Nicholas L. Mills [27,28] &
Hüseyin Firat[21]

Tools for predicting COVID-19 outcomes enable personalized healthcare, potentially easing the disease burden. This collaborative study by 15 institutions across Europe aimed to develop a machine learning model for predicting the risk of in-hospital mortality post-SARS-CoV-2 infection. Blood samples and clinical data from 1286 COVID-19 patients collected from 2020 to 2023 across four cohorts in Europe and Canada were analyzed, with 2906 long non-coding RNAs profiled using targeted sequencing. From a discovery cohort combining three European cohorts and 804 patients, age and the long non-coding RNA LEF1-AS1 were identified as predictive features, yielding an AUC of 0.83 (95% CI 0.82–0.84) and a balanced accuracy of 0.78 (95% CI 0.77–0.79) with a feedforward neural network classifier. Validation in an independent Canadian cohort of 482 patients showed consistent performance. Cox regression analysis indicated that higher levels of LEF1-AS1 correlated with reduced mortality risk (age-adjusted hazard ratio 0.54, 95% CI 0.40–0.74). Quantitative PCR validated LEF1-AS1's adaptability to be measured in hospital settings. Here, we demonstrate a promising predictive model for enhancing COVID-19 patient management.

On October 2nd, 2023, the Nobel Assembly at Karolinska Institute awarded the 2023 Nobel Prize in Physiology or Medicine to Professors Katalin Karikó and Drew Weissman for their discovery that modifying the uridine nucleoside to pseudouridine blocks the inflammatory response consecutive to cell delivery of messenger RNA (mRNA) molecules, thereby increasing the production of proteins encoded by the mRNA[1]. This discovery 15 years ago revolutionized the therapeutic potential of mRNA and allowed the rapid development of mRNA vaccines against SARS-CoV-2. RNAs have come of age, not only for vaccines, but for diagnosing and treating disease[2].

On March 2020, partners of the EU-CardioRNA COST Action network[3,4] gathered forces to develop a RNA-based diagnostic test

based on artificial intelligence (AI) to predict clinical outcomes after COVID-19[5]. The rationale for this endeavor was that leveraging the power of non-coding RNAs may help reduce the devastating consequences of COVID-19 pandemic[6]. Indeed, risk prediction models could inform about clinical management of patients. Non-coding RNAs, unable to encode proteins like the better-known mRNAs, are regulated in virtually all pathological conditions and, since they are detectable in the blood, they have emerged in recent years as a new reservoir of non-invasive candidate biomarkers and therapeutic targets. Our consortium previously characterized a panel of 2906 cardiac-enriched or heart failure-associated long non-coding RNAs (lncRNAs) (FIMICS panel)[7] which, together with an in-house developed bioinformatics pipeline to maximize the benefit of targeted sequencing (Firalink pipeline[8]), provides a new tool to discover disease-associated lncRNAs with potential to help in diagnosis and risk stratification. Since the FIMICS panel contains many inflammation-related lncRNAs and inflammation is a hallmark of host response to infection by SARS-CoV-2, we thought that it may be of usefulness to identify predictors of COVID-19 outcome.

In the H2020-funded FastTrack COVIRNA project, we aimed to apply the FIMICS panel to identify lncRNAs predictive of COVID-19 outcome. We used blood samples and clinical data from four cohorts of COVID-19 patients totaling 1286 patients. Three cohorts with 804 patients were merged as a discovery cohort for feature selection and choice of best performing machine learning (ML) models. The fourth cohort of 482 patients was used for validation purposes. Here, we have built a model based on one lncRNA and age able to predict in-hospital mortality with an area under the receiver operating characteristic curve (AUC) of 0.83 (0.82–0.84).

## Results
### Study design
The study design is illustrated in Fig. 1. The study population consisted of a total of 1329 patients with COVID-19, shared between a discovery cohort ($n = 818$) and a validation cohort ($n = 511$) used for ML model selection and evaluation, respectively. Three European cohorts were included in the discovery cohort (PrediCOVID from Luxembourg, $n = 141$; NAPKON from Germany, $n = 557$; and ISARIC4C from United Kingdom, $n = 120$) and one cohort from Canada constituted the validation cohort (BQC19, $n = 511$). Whole blood samples collected in PAXgene RNA tubes at baseline in all patients were centrally stored at −80 °C

in a NF S96-900 certified biobank. RNA extraction, quality check, library preparation and RNA sequencing using the FIMICS panel were performed in our core lab. Raw sequencing data were normalized and merged with clinical data of patients in our central database. Data were curated and made available for analysis using ML/AI. Patients with RNAseq datasets that did not meet the quality criteria described in the Materials and Methods section, or with blood samples not collected at the time of enrolment in the study, or for which survival data were not available, were excluded from the analysis. After curation and quality checks, combined RNAseq datasets and clinical data from 136 PrediCOVID, 556 NAPKON, 112 ISARIC4C (804 patients for the discovery cohort) and 482 BQC19 patients for the validation cohort were available for ML analysis. Overall, a total of 1286 full datasets representing each a unique patient were available for analysis. After lncRNA selection by ML, a translational study was conducted by qPCR in a subgroup of 86 patients from the NAPKON cohort for which leftover RNA was available.

Baseline characteristics of patients in the analysis are reported in Table 1, in which the three merged European cohorts used for discovery are compared to the Canadian cohort used for validation of the selected features and ML models. Missing data are indicated and were imputed using missForest. The median number of days in hospital was 9 (Q1 = 5, Q3 = 19) and 8 (Q1 = 4, Q3 = 19) for the ISARIC4 and BQC19 cohort, respectively. In all cohorts, patients who died in hospital were older than survivors, more often had cardiovascular disease, and more often received oxygen therapy. Being a male was associated with a higher risk of death in the merged European cohorts. Diabetes and chronic lung disease were also risk factors in this cohort. Patients in the Canadian cohort were older, were more often females and were less often smokers than patients in the merged European cohorts. Supplementary Table 1 shows the characteristics of the three European cohorts individually, together with the nature of common COVID-19 symptoms across cohorts. The PrediCOVID cohort had younger patients than the two other cohorts and none of them died during the follow-up period. There were more smokers at the time of enrolment in the PrediCOVID cohort than in the NAPKON and ISARIC4C cohorts. Common baseline symptoms across the European cohorts included fever, headache, cough and dyspnea, which were less frequent in survivors (Supplementary Table 1). Ethnicity data was available in the NAPKON cohort, in which most patients were Caucasian and no apparent association between ethnicity and survival was found (Supplementary Table 1). In this cohort, vaccinated people had a lower risk

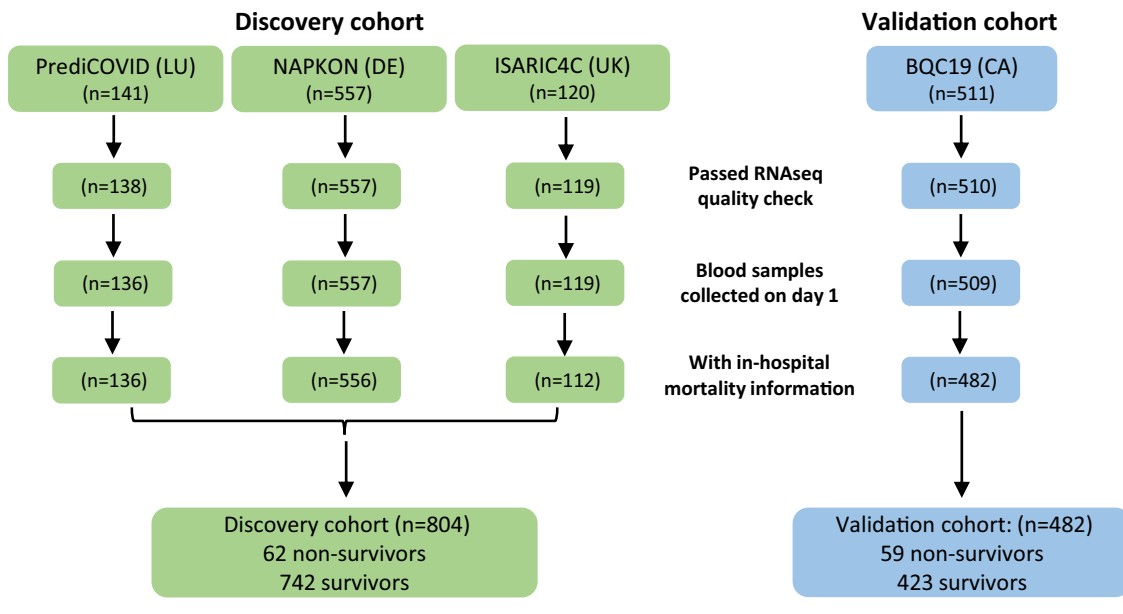

**Fig. 1 | Study design.**

**Table 1 | Baseline characteristics of patients in the discovery and validation cohorts**

| | Discovery cohort (PrediCOVID+NAPKON + ISARIC4C; n = 804) | | | | | Validation cohort (BQC19; n = 482) | | | | | P discovery vs. validation |
|---|---|---|---|---|---|---|---|---|---|---|---|
| | All | Non-survivors | Survivors | Missing data | P non-survivors vs. survivors | All | Non-survivors | Survivors | Missing data | P non-survivors vs. survivors | |
| Age (years, mean [SD]) | 54 [16] | 69 [11] | 52 [16] | 4 | 1.0E-16 | 65 [18] | 79 [12] | 63 [17] | 0 | 7.5E-15 | 2.4E-28 |
| Male (n [%]) | 480 [59.7] | 46 [74.2] | 434 [58.5] | 0 | 1.5E-02 | 259 [53.7] | 32 [54.2] | 227 [53.7] | 0 | 1.0E+00 | 4.1E-02 |
| Current smoker (n [%]) | 73 [11.2] | 3 [6.4] | 70 [11.5] | 150 | 3.4E-01 | 19 [4.6] | 2 [4.4] | 17 [4.6] | 68 | 1.0E+00 | 1.3E-04 |
| Former smoker (n [%]) | 205 [31.3] | 18 [38.3] | 187 [30.8] | 150 | 3.3E-01 | 88 [21.3] | 12 [26.7] | 76 [20.6] | 68 | 3.4E-01 | 3.2E-04 |
| Diabetes (n [%]) | 152 [19.29] | 22 [36.07] | 130 [17.88] | 16 | 1.2E-03 | 154 [32.02] | 25 [42.37] | 129 [30.57] | 1 | 7.5E-02 | 3.7E-07 |
| Chronic lung disease (n [%]) | 136 [17.24] | 18 [30.00] | 118 [16.19] | 15 | 1.2E-02 | 101 [21.22] | 15 [25.86] | 86 [20.57] | 6 | 3.9E-01 | 8.7E-02 |
| Cardiovascular disease (n [%]) | 298 [37.96] | 39 [65.00] | 259 [35.72] | 19 | 1.2E-05 | 292 [60.71] | 51 [87.93] | 241 [56.97] | 1 | 2.5E-06 | 3.6E-15 |
| Oxygen therapy (n [%]) | 431 [54.42] | 56 [91.80] | 375 [51.30] | 12 | 8.2E-11 | 329 [75.46] | 56 [94.92] | 273 [72.41] | 46 | 6.2E-05 | 2.0E-13 |

The p values for continuous variables are from 2-sided Student's t test. The p values for categorical variables are from 2-sided Fisher exact test.

of death as compared to non-vaccinated people (Supplementary Table 1). Vaccination data was unavailable in other cohorts.

### Machine learning model building and characterization

We performed feature selection on the training set and evaluated five different ML classifiers (RF, kNN, Logit, MLP, SVM, XGB) on the discovery cohort derived from the 3 combined European cohorts (n = 804) in each of the 100 iterations as described in the Materials and Methods section and in Supplementary Fig. 1. The median number of features selected in each iteration was 21 (Q1 = 16, Q3 = 25). The performance of each model to predict in-hospital mortality is shown in Table 2. The logistic regression model (Logit) with the selected features in each iteration provided the most accurate prediction of in-hospital mortality with an AUC of 0.83 (95% CI 0.81–0.84), an accuracy of 0.74 (95% CI 0.73–0.76), a sensitivity of 0.77 (95% CI 0.74–0.79), and a specificity of 0.72 (95% CI 0.69–0.75).

The analysis yielded the selection of two features, age and the lncRNA LEF1-AS1, which appeared in 82 and 63 iterations out of the 100 iterations performed, respectively (Fig. 2A). LEF1-AS1 is a lncRNA of 3,360 nucleotides transcribed from the lymphoid enhancer binding factor 1 (LEF1) locus located on chromosome 4. In the merged European cohorts (discovery cohort, n = 804), patients who survived were younger and had higher expression levels of LEF1-AS1 than patients who died (Fig. 2B, C). There was a significant albeit moderate negative correlation between age and LEF1-AS1 in this cohort (Fig. 2D), as well as in the Validation cohort (r = −0.35, p < 0.01). Also, LEF1-AS1 was differentially expressed between males and females in the Discovery (Fig. 2E) and in the Validation cohort (p < 0.01 and p = 0.02, respectively). The expression of LEF1-AS1 was associated with cancer diagnosis with an odds ratio of 0.71 [0.55–0.90] and 0.66 [0.52–0.84] in the NAPKON and BQC19 cohorts, respectively. The Shapley beeswarm plots shown in Supplementary Fig. 2 attest that higher age and lower expression of LEF1-AS1 led to positive SHAP values and thus had positive impacts on model output.

The five different ML classifiers with the two selected features (age and LEF1-AS1) were then evaluated on the discovery cohort in 100 iterations, using the same data splits as for feature selection. The model MLP exhibited the highest AUC of 0.82 (95% CI: 0.80–0.84) (Table 3 and Supplementary Fig. 3). There was no significant difference in performance between the models with the features from each iteration and the model with age and LEF1-AS1 (Tables 2 and 3). Adding the third best predictor selected during the feature selection step did not improve the performance of the prediction model in the balanced (AUC 0.84 [0.82–0.86]. p = 0.11 for comparison with the model without oxygen therapy) and imbalanced (AUC = 0.83 [0.82–0.84], p = 0.91 for comparison with the model without oxygen therapy) discovery dataset.

When predicting in-hospital mortality for the balanced datasets from the validation cohort (i.e., same number of survivors and deceased patients, Supplementary Fig. 1), the MLP model achieved an AUC of 0.84 (95% CI 0.82–0.86), an accuracy of 0.76 (95% CI 0.74–0.78), a sensitivity of 0.77 (95% CI 0.75–0.79), and a specificity of 0.75 (95% CI 0.72–0.78) (Table 4). We extended the testing to the original imbalanced datasets using the 2 selected features, yielding the following metrics for the discovery cohort: AUC 0.83 (95% CI 0.82–0.84), balanced accuracy 0.78 (95% CI 0.77–0.79), sensitivity 0.86 (95% CI 0.84–0.88), and specificity 0.71 (95% CI 0.70–0.71); for the validation cohort, the metrics were AUC 0.83 (95% CI 0.82–0.84), balanced accuracy 0.75 (95% CI 0.74–0.77), sensitivity 0.79 (95% CI 0.76–0.82), and specificity 0.72 (95% CI 0.71–0.73). The model with age alone yielded AUC 0.78 (95% CI 0.77–0.80), 0.79 (95% CI 0.78–0.80), 0.78 (95% CI 0.76–0.79), and 0.78 (95% CI 0.76–0.79) in balanced/ imbalanced discovery and balanced/imbalanced validation cohort, respectively. Adding LEF1-AS1 significantly improved the model performance (Fig. 3, Supplementary Fig. 4). Adding sex and/or the other 2 features which were selected more than 40 times in the feature

**Table 2 | Performance of different classifiers to predict in-hospital mortality in the discovery cohort (*n* = 804) using the features from each iteration**

| Classifier | AUC (95% CI) | Accuracy (95% CI) | Sensitivity (95% CI) | Specificity (95% CI) | Brier score (95% CI) |
|---|---|---|---|---|---|
| RF | 0.81 (0.8–0.83) | 0.73 (0.71–0.75) | 0.74 (0.72–0.76) | 0.73 (0.7–0.76) | 0.18 (0.17–0.19) |
| kNN | 0.82 (0.8–0.84) | 0.74 (0.72–0.76) | 0.74 (0.71–0.76) | 0.74 (0.71–0.77) | 0.18 (0.17–0.19) |
| Logit | 0.83 (0.81–0.84) | 0.74 (0.73–0.76) | 0.77 (0.74–0.79) | 0.72 (0.69–0.75) | 0.18 (0.17–0.19) |
| MLP | 0.81 (0.8–0.83) | 0.73 (0.71–0.75) | 0.76 (0.73–0.78) | 0.71 (0.68–0.73) | 0.19 (0.18–0.20) |
| SVM | 0.76 (0.73–0.8) | 0.75 (0.73–0.76) | 0.78 (0.75–0.8) | 0.72 (0.69–0.74) | 0.19 (0.18–0.20) |
| XGB | 0.74 (0.72–0.76) | 0.67 (0.65–0.69) | 0.7 (0.67–0.72) | 0.64 (0.61–0.67) | 0.26 (0.24–0.27) |

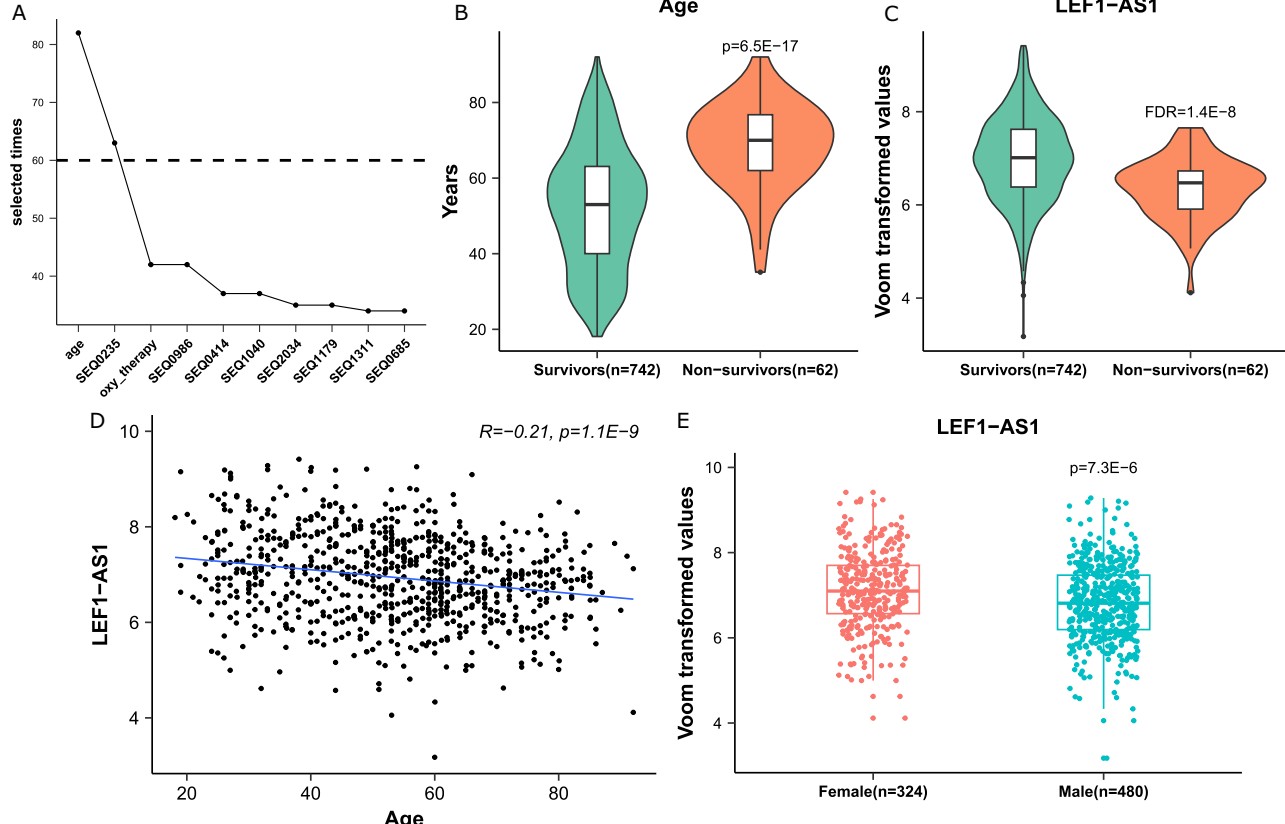

**Fig. 2 | Feature selection on the discovery cohort (*n* = 804 patients). A** Line plot of the selected times of the 10 most selected features. *X*-axis: the name of the features. SEQXXXX are the codes of the probes of the FIMICS panel. SEQ0235 probe recognizes the lncRNA LEF1-AS1. *Y*-axis: the number of times a feature appeared in the 100 iterations of the feature selection process. **B**, **C** Box/violin plots of age and LEF1-AS1 expression, which were significantly increased and decreased in the non-survivors group (*n* = 62 patients) of the European cohorts, respectively. *P*-value is from 2 sided Student's *t* test. FDR (false discovery rate) is from DESeq2 algorithm. **D** Correlation between age and LEF1-AS1. A Pearson Correlation coefficient and a two-sided *t*-test *p*-value are indicated. **E** Comparison between expression levels of LEF1-AS1 in males (*n* = 480 patients) and females (*n* = 324 patients). *P*-value is from a two-sided Student's *t* test. In **B**, **C** and **E**, the box is drawn from Q1 (25th percentile) to Q3 (75th percentile) with a horizontal line inside it to denote the median. The length of the whiskers indicate 1.5 times of IQR (Interquartile range Q3–Q1).

**Table 3 | Performance of different classifiers to predict in-hospital mortality in the discovery cohort using the two selected features (age and LEF1-AS1)**

| Classifier | AUC (95% CI) | Accuracy (95% CI) | Sensitivity (95% CI) | Specificity (95% CI) | Brier score (95% CI) |
|---|---|---|---|---|---|
| RF | 0.78 (0.76–0.80) | 0.73 (0.71–0.75) | 0.76 (0.73–0.78) | 0.71 (0.68–0.74) | 0.19 (0.18–0.20) |
| kNN | 0.81 (0.79–0.83) | 0.75 (0.73–0.77) | 0.85 (0.83–0.87) | 0.65 (0.62–0.68) | 0.18 (0.17–0.19) |
| Logit | 0.81 (0.79–0.83) | 0.76 (0.74–0.77) | 0.81 (0.78–0.84) | 0.70 (0.67–0.73) | 0.18 (0.17–0.19) |
| MLP | 0.82 (0.80–0.84) | 0.77 (0.75–0.79) | 0.82 (0.80–0.84) | 0.72 (0.69–0.75) | 0.18 (0.17–0.18) |
| SVM | 0.67 (0.62–0.72) | 0.74 (0.72–0.76) | 0.82 (0.80–0.84) | 0.67 (0.63–0.70) | 0.21 (0.20–0.22) |
| XGB | 0.74 (0.72–0.76) | 0.68 (0.66–0.70) | 0.69 (0.66–0.71) | 0.67 (0.64–0.70) | 0.25 (0.24–0.26) |

**Table 4 | Performance of the MLP model to predict in-hospital mortality in the balanced/imbalanced discovery and validation cohorts**

| Dataset | | AUC (95% CI) | Balanced accuracy (95% CI) | Sensitivity (95% CI) | Specificity (95% CI) | Brier score (95% CI) |
|---|---|---|---|---|---|---|
| Balanced | Validation cohort | 0.84 (0.82–0.86) | 0.76 (0.74–0.78) | 0.77 (0.75–0.79) | 0.75 (0.72–0.78) | 0.17 (0.16–0.18) |
| Imbalanced | Discovery cohort | 0.83 (0.82–0.84) | 0.78 (0.77–0.79) | 0.86 (0.84–0.88) | 0.71 (0.70–0.71) | 0.18 (0.18–0.18) |
| | Validation cohort | 0.83 (0.82–0.84) | 0.75 (0.74–0.77) | 0.79 (0.76–0.82) | 0.72 (0.71–0.73) | 0.18 (0.18–0.18) |

Balanced accuracy means accuracy for balanced data.

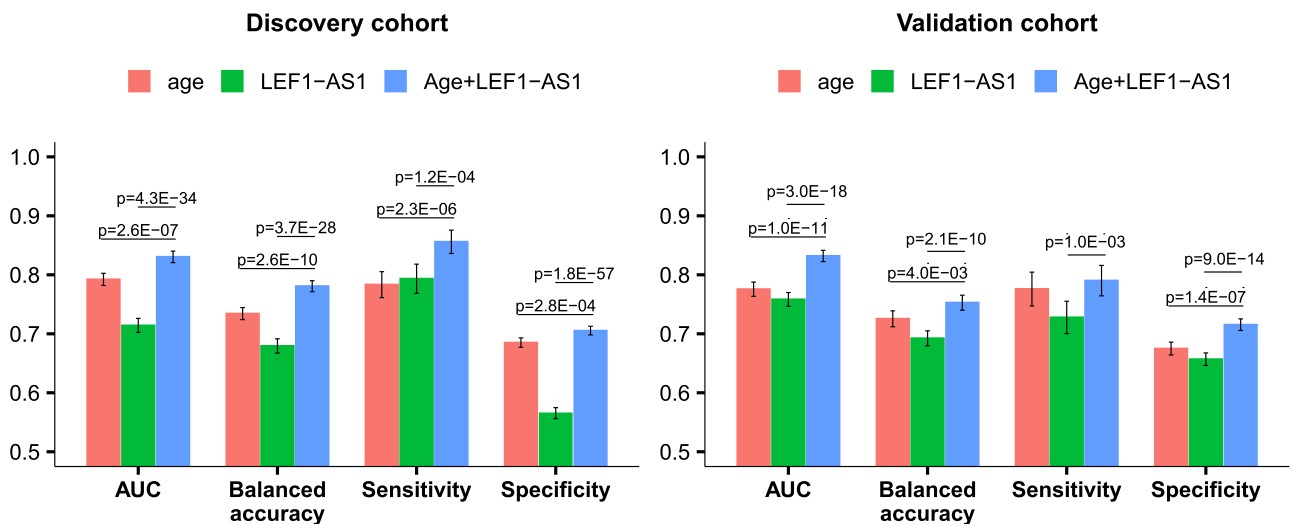

**Fig. 3 | Comparison of the performance of the models with age alone, LEF1-AS1 alone and the two features using the discovery (*n* = 804 patients) and the validation cohort (*n* = 482 patients), respectively.** The evaluation was performed on the imbalanced data with 20 repeated 5-fold cross-validation. The error bars display the confidence interval. We indicated the significant (*p* < 0.05) difference compared to the model with 2 features using a two-sided Student's *t* test.

selection iterations (oxygen therapy and SEQ0986, Fig. 2A) did not significantly improve the model performance (Supplementary Fig. 5). Missing data imputation did not significantly influence results since the MLP model run without prior imputation of missing age data (concerning only 4 patients of the Discovery cohort, Table 1) reached an AUC of 0.83 (95% CI 0.81–0.85) and a balanced accuracy of 0.78 (95% CI 0.76–0.80) (Supplementary Table 3).

We compared the predictive performance of the MLP model with age and LEF1-AS1 to a previously published model involving age, sex, C-reactive protein (CRP) and lactate dehydrogenase (LDH). As shown in Supplementary Fig. 6, our MLP model and the four-parameter model had similar capacity to predict mortality in the BQC19 cohort (AUC 0.83 [0.82–0.84] vs 0.85 [0.84–0.86], respectively). Our MLP model outperformed the four-parameter model in the NAPKON cohort (AUC 0.82 [0.81–0.83] vs 0.78 [0.76–0.79], respectively). Brier score analysis was used to assess the calibration of our MLP model, where a lower Brier score indicates a more calibrated model. This analysis revealed a similar (for BQC-19 data) and a lower score (for NAPKON data) for our MLP model compared to the previously published four-parameter model (Supplementary Fig. 6).

#### Survival analysis

We then evaluated the association between the lncRNA LEF1-AS1 and in-hospital mortality using survival analysis. Patients with high levels of LEF1-AS1 were at low risk of death (age-adjusted HR 0.59, 95% CI 0.36–0.96) in the ISARIC4C subgroup of the discovery cohort (Fig. 4A). In the validation cohort, the HR was 0.54 (95% CI 0.40–0.74) (Fig. 5A). Kaplan–Meier curves using different cut-offs for LEF1-AS1 expression demonstrate the observed association of high expression levels of LEF1-AS1 with low risk of death (Figs. 4B and 5B).

#### Translational perspective

To gain further insights into the feasibility of LEF1-AS1 testing in the hospital environment, e.g., for the development of a molecular diagnostic assay, we set-up a quantitative PCR protocol to measure blood levels of LEF1-AS1 in a subgroup of 84 patients of the NAPKON cohort. Patient characteristics are shown in Supplementary Table 2. 41 patients survived and 43 died in hospital. The two groups were age-matched, sex-balanced and had similar average body mass index (BMI). We first validated that expression levels of LEF1-AS1 as assessed by quantitative PCR were correlated with the levels obtained by RNAseq using the FIMICS panel (Fig. 6A). Moreover, as shown in Fig. 6B, patients who died during their hospital stay had a lower expression of LEF1-AS1 compared to survivors (*p* = 0.003). A patient was -1.4 times as likely to survive at hospital discharge for every 1 unit (log2 transformed expression) increase in LEF1-AS1 (OR 1.39 95% CI 1.10–1.76). When we dichotomized the log2-transformed expression levels of LEF1-AS1 using a cut-off determined by the Youden's index (to maximize specificity and sensitivity), patients who had LEF1-AS1 levels above 0.043 were 5 times more likely to survive after hospital discharge (OR 5.08 95% CI 2.02–12.73).

#### Discussion

We hereby report the characterization of a machine learning model based on age and the lncRNA LEF1-AS1 able to predict in-hospital mortality of COVID-19 patients with clinically relevant accuracy.

COVID-19 pandemic has impacted peoples' lives in many different ways. Healthcare management during the pandemic has been challenging, partly due to lack of preparedness and ability to triage the large numbers of people with infection arriving at the Emergency Department. Methods to help triage and risk stratify patients would have greatly facilitated the work of healthcare providers. Being able to

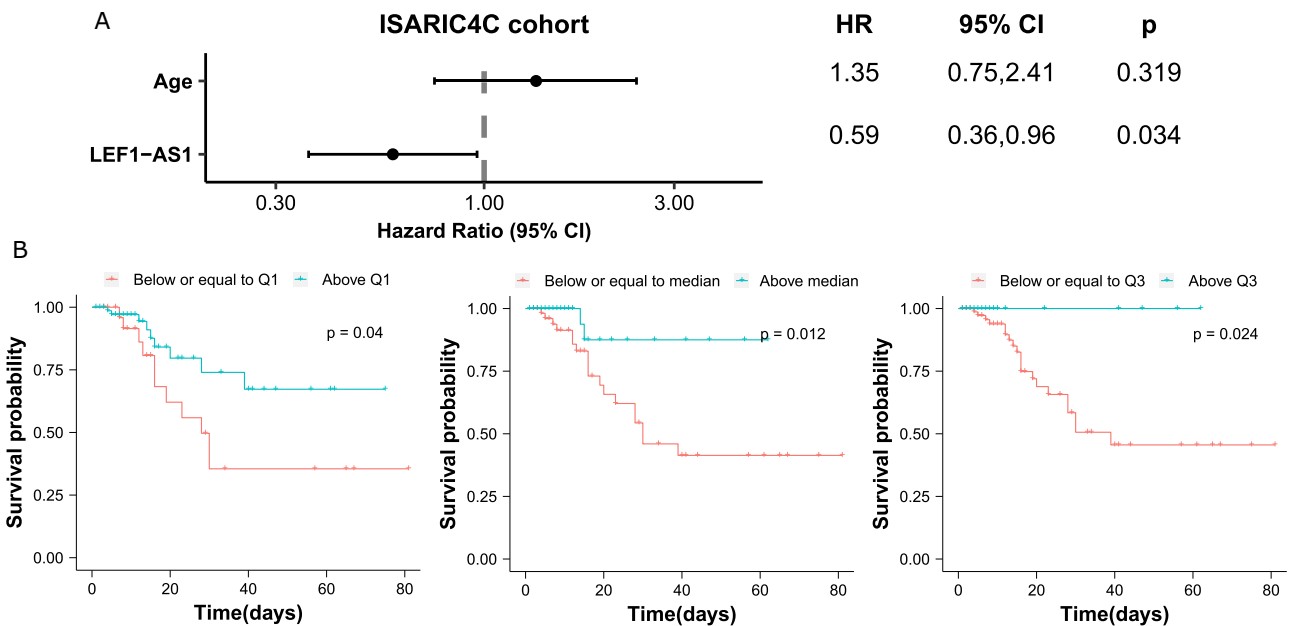

**Fig. 4 | Survival analysis in the ISARIC4C cohort (*n* = 112 patients). A** Forest plot of the Hazard Ratio (HR) from Cox regression analysis shows a higher risk of death for older patients and a lower risk for patients with higher LEF1-AS1 expression level. The dots and the error bars display the HR and the confidence interval, respectively. The *p* values are from a two-sided Wald test. **B** Kaplan–Meier curves using the stratified LEF1-AS1 expression with the first quartile (Q1), the median and the third quartile (Q3), respectively. Patients with LEF1-AS1 expression levels below or equal to the first quartile (Q1) are at a high risk of death. The *p* values are from a two-sided log-rank test.

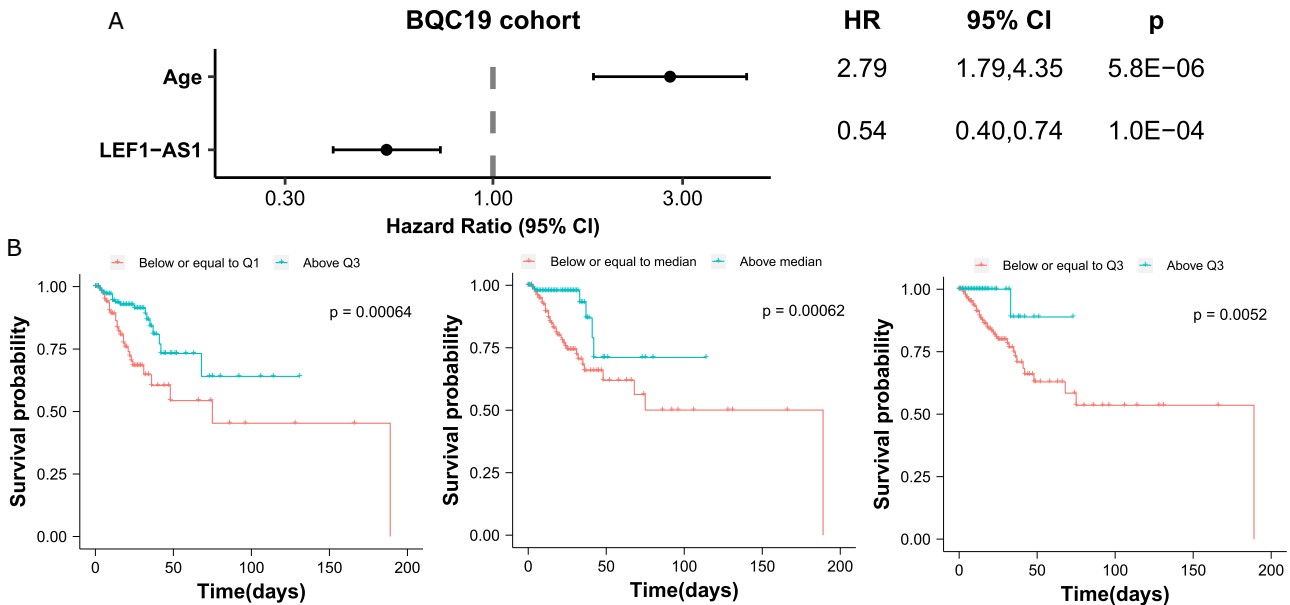

**Fig. 5 | Survival analysis in the BQC19 cohort (*n* = 438 patients).** Note that 44 patients of the 482 patients of the BQC19 cohort did not have information on the number of hospitalized days, yet had information on in-hospital mortality. **A** Forest plot of the Hazard Ratio (HR) from Cox regression analysis shows a higher risk of death for older patients and a lower risk for patients with higher LEF1-AS1 expression level. The dots and the error bars display the HR and the confidence interval, respectively. The *p* values are from a two-sided Wald test. **B** Kaplan–Meier curves using the stratified LEF1-AS1 expression with the first quartile (Q1), the median and the third quartile (Q3), respectively. Patients with LEF1-AS1 expression levels below or equal to the first quartile (Q1) are at a high risk of death. The *p* values are from a two-sided log-rank test.

identify patients at high-risk of poor outcome or death, or on the other hand patients with a high chance of survival, would have allowed a more personalized approach to the use of healthcare that could have improved outcomes overall.

Initiated in March 2020 during the first phase of the pandemic, this study aimed to cope with the above issue and design a new method to identify patients at high risk of poor outcome after being

infected with SARS-CoV-2. We applied our previously developed FIMICS panel of lncRNAs[7] to whole blood samples of COVID-19 patients collected from four different European cohorts and a Canadian cohort. This panel allows for targeted sequencing, which is about 70 times more sensitive than whole genome sequencing, and therefore more suitable to detect and quantify potentially weakly expressed lncRNAs. Other studies have identified biomarkers of disease severity

A

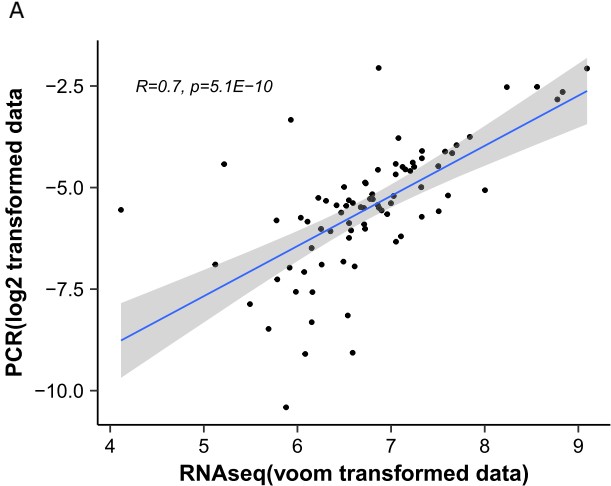

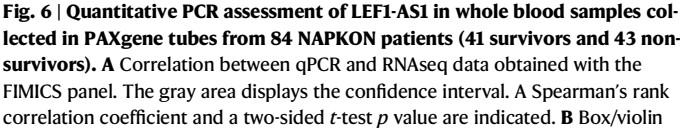

B

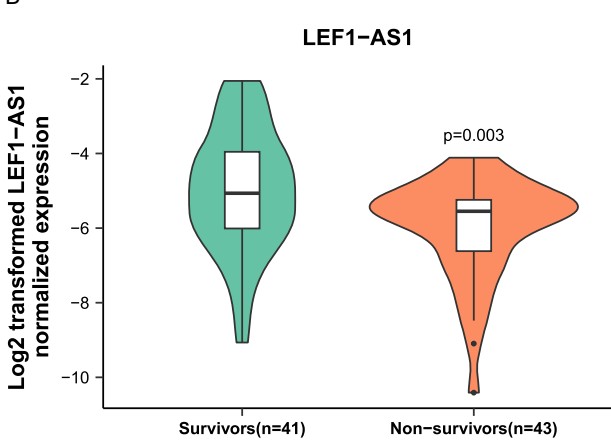

**Fig. 6 | Quantitative PCR assessment of LEF1-AS1 in whole blood samples collected in PAXgene tubes from 84 NAPKON patients (41 survivors and 43 non-survivors). A** Correlation between qPCR and RNAseq data obtained with the FIMICS panel. The gray area displays the confidence interval. A Spearman's rank correlation coefficient and a two-sided *t*-test *p* value are indicated. **B** Box/violin plots of LEF1-AS1 expression, which was decreased in deceased patients (*n* = 43 patients). The box is drawn from Q1 (25th percentile) to Q3 (75th percentile) with a horizontal line inside it to denote the median. The length of the whiskers indicate 1.5 times IQR (Interquartile range Q3−Q1). *P*-value is from a two-sided Student's *t*-test.

and outcome of COVID-19[9–11]. We previously reported that LEF1-AS1 expression in peripheral blood cells was negatively associated with disease severity and mortality in a modestly sized cohort of COVID-19 patients[12], which is consistent with our present investigation in whole blood samples. Models to predict mortality of COVID-19 patients have been previously developed, yet they suffer from a high risk of bias[13]. The MLP model reported in the present study with only two features (age and LEF1-AS1) showed similar predictive performance in the BQC-19 cohort and higher performance in the NAPKON cohort compared to a model including age, sex, CRP and LDH. As compared to previous reports[13], the strength of our study relies on its methodological aspects which reduce the risks of bias. We conducted a multi-center and well powered study, with patient numbers well above previous studies. We have used a machine learning pipeline including feature selection and testing of multiple machine learning models with Discovery and Validation cohorts, each split into training and testing subgroups. In each cohort, we have evaluated models on the imbalanced datasets using twenty times repeated 5-fold cross validation.

Even though we observed a consistently low expression of LEF1-AS1 in patients with high risk of death, a functional role of LEF1-AS1 in COVID-19 outcome has still to be demonstrated. LEF1-AS1 is an antisense RNA to the lymphoid enhancer binding factor 1 (LEF1) gene encoding a transcription factor expressed in pre-B and T cells which is involved in proliferation, activation of genes in the Wnt/β-catenin pathway and in regulating systemic inflammation. Consistent with our observed lower expression of LEF1-AS1 in severe patients, recent studies have illustrated that B cells undergo significant depletion following SARS-CoV-2 infection[14]. Additionally, pulmonary fibrosis stems from damage to alveoli and is a hallmark of SARS-CoV-2 infection. Recent work has demonstrated that alveolar damage can be suppressed through activation of LEF1, which is mediated by the transcription factor krüppel-like factor 4, thus hinting at a possible protective role of LEF1 following alveolar injury and SARS-CoV-2 infection[15]. These studies suggest a link between LEF1/LEF1-AS1, T or B cell proliferation, alveolar protection and COVID-19 severity which warrants further investigation.

The machine learning protocol used in the present study was inspired by the method from ref. 16, which used Boruta, a random forest-based algorithm, to select features from electronic health records and evaluate a quantitative marker of coronary artery disease. We adapted their design to suit RNAseq data by adding DESeq2 for differential expression analysis. Many conventional statistical

methods, such as t-tests and ANOVA, assume normal data distributions, which is often not the case for data generated by high throughput platforms, such as sequencing. New machine learning methods are able to deal with scale, diverse data distributions, and non-linearity, such as large omics datasets[17]. Multiple machine learning algorithms, including deep learning algorithms, have been developed to build powerful predictive models linking omics data to prediction of clinical outcomes[18,19]. While benefiting from the modeling flexibility and robustness, these models often suffer from difficulty in interpreting the role of each individual feature. Identifying biomarkers functionally associated with disease progression could help establish novel hypotheses regarding prevention, diagnosis, and treatment of complex human diseases[20].

**Translational perspectives**

The present investigation was conducted using patient's whole blood samples collected in PAXgene RNA tubes, which are certified for in vitro diagnostics. Other matrices could also be used and we do not exclude that other biomarkers may be found with relevant predictive value. Using quantitative PCR, a technique available in most hospital labs and cost-effective, we confirmed that LEF1-AS1 was readily and reliably detected. Furthermore, we validated that low levels of LEF1-AS1 were associated with a high risk of death. These data support the potential translation of our findings to clinical application.

With the current excitement for the use of RNA as both biomarker and therapeutic targets, lncRNAs may constitute a novel generation of actionable disease-monitoring biomarkers and drugs. Our data showing that lncRNAs are associated with mortality of COVID-19 patients support their potential as theranostic drugs, usable for both risk assessment and treatment of COVID-19. Circular RNAs particularly raised interest for future drug development since these closed RNA molecules are not only able to more stably induce therapeutic protein production compared to linear RNAs, they also have potential to capture and sequester unwanted molecules and thereby function as antisense RNAs, or they can regulate RNA editing[21]. Whether lncRNAs find utility for COVID-19 remains to be determined, as well as whether circRNAs hold similar or superior value to reduce disease burden. It will be interesting in such endeavors to develop multimodal approaches taking into account not only baseline clinical characteristics and biomarkers but also mental health indicators, considering the importance of pre-existing health problems and especially psychological

problems in the development of post-COVID condition[22]. It would be interesting to apply a similar approach to see whether lncRNAs are associated with the long term impact of COVID-19, such as long COVID[23]. Considering the prevalence and devastating consequences of this novel disease[24], setting-up methods to predict the risk of developing long COVID symptoms would have a significant impact on the enormous burden of long COVID or post-COVID symptoms.

## Limitations

This work has some limitations. First, since patients enrolled in this study were from the first phase of the pandemic, we assume that most if not all patients were infected by the original SARS-CoV-2 variant. Also, there was limited information on vaccination status due to the fact that there were no widely available vaccines at the time of study enrolment. However, we cannot exclude that some patients were infected by other variants. Hence, we could not test the performance of the model in patients infected by different viral variants. Second, only limited clinical descriptions of the patients enrolled in the study could be provided due to heterogeneity of cohorts and difficulty to merge the clinical data from different cohorts. Third, none of the participants of the Luxembourg PrediCOVID cohort died in hospital, most probably due to the nationwide mass screening program, which allowed an improved control of the virus and an earlier hospitalization of patients[25]. Since this cohort was included at project inception and despite that the main aim of this study was to predict in-hospital mortality, it was kept in analyses and we verified that its removal does not affect study findings. Fourth, survival analysis using Cox regression and Kaplan–Meier curves could be conducted only in the ISARIC4C and BQC19 subgroups for which we had data on time to death. Fifth, even though we tested five different ML classifiers, others could provide stronger predictive value. Lastly, a full functional characterization of the role of LEF1-AS1 in post COVID-19 outcome remains to be done. We identified a machine learning-supported model combining age and the lncRNA LEF1-AS1 predictive of COVID-19 in-hospital mortality. This model may find utility for the management of COVID-19 patients. Its usefulness for long COVID patients remains to be tested.

## Methods

### Patient cohorts

This study was performed in full compliance with the Declaration of Helsinki. Involved cohorts comprise COVID-19-positive patients aged 18 years and older from Luxembourg (PrediCOVID study), Germany (NAPKON study), United Kingdom (ISARIC4C study), and Canada (BQC19 study). The Luxembourg PrediCOVID study was approved by the National Research Ethics Committee of Luxembourg (study Number 202003/07) and was registered under ClinicalTrials.gov (NCT04380987)[26]. The ISARIC-4C study was approved by the Oxford C Research Ethics Committee (Reference 13/SC/0149) (details on study design, registration and approvals are available in the online supplement). For the NAPKON Cross-Sectoral Platform, a primary ethics vote was obtained at the Ethics Committee of the Department of Medicine at Goethe University Frankfurt, Germany (local ethics ID approval 20-924). All further study sites received their local ethics votes at the respective ethics committees. The NAPKON Cross-Sectoral Platform is registered at ClinicalTrials.gov (Identifier: NCT04768998)[27]. The Biobanque québécoise de la COVID-19 (BQC19) study has been approved by the Research Ethics Board of the Center Hospitalier de l'Université de Montréal (CHUM) (#13.389)[28]. Periods of patient enrolment and biological samples collection were as follows: May 2020 - Present for PrediCOVID, July 2020 - Present for NAPKON, February 2020 - September 2020 for ISARIC4C, March 2020 - Present for BQC19. Informed consent was signed by all patients enrolled in these studies. Legal agreements for material and data sharing have been signed between each cohort and COVIRNA project coordinator Luxembourg Institute of Health (LIH).

### Sample storage and RNA extraction

All procedures were performed in the ISO 17025, ISO 9001, and CAP accredited facility of Firalis. Whole blood samples collected in PAXgene™ Blood RNA tubes (PreAnalytiX, Cat. #762165; BD Biosciences, Aalst, Belgium) were shipped from the different patient cohorts to our central NF S96-900 certified Biobank and were stored at −80 °C. Whole blood samples were randomized according to age and sex in batches of 64 prior to RNA extraction. Total RNA was extracted with the KingFisher Apex instrument (Cat. #5400930 P, Thermo Scientific, Waltham, MA, USA) using the MagMAX™ for Stabilized Blood Tubes RNA Isolation Kit (Cat. #4451894, Invitrogen, Thermo Scientific). Extracted RNA samples were quantified using the Qubit 3.0 fluorometer (Cat. #Q33216, Invitrogen, Thermo Fisher Scientific) with the RNA high sensitivity Assay kit. Sample quality was assessed using a TapeStation 4150 electrophoresis platform (Cat. #G2992AA, Agilent, Santa Clara, CA, USA).

### Library preparation, targeted RNA sequencing and raw data analysis

An extended version of this section is available in the Supplementary Material. Briefly, a second stratified randomization by age and sex was performed in batches of 46 samples prior to library preparation. The libraries were generated by the EpMotion 5075t NGS solution (Cat. #5075000962, Eppendorf, Hamburg, Germany) using the KAPA Stranded RNAseq Kit with RiboErase (HMR; Cat. #634444, Roche diagnostics, Basel, Switzerland) for ribosomal RNA (rRNA) depletion and total RNA libraries construction. The clean-ups were performed with Celemag clean-up beads (Cat. #CMCB57.6, Celemics, Seoul, Korea) and the purified libraries were dual indexed during a 13-cycle PCR using the library preparation box #2 (Cat. # LI20D96, Celemics).

The indexed libraries were then captured using the FIMICS panel targeting 2906 lncRNAs[7] (Cat. #BO5096, Celemics) and purified using Celemag streptavidin coated magnetic beads (Cat. #CMSB5.76, Celemics) and Celemics wash buffer (Cat. #TC4096, Celemics). The on-beads captured sequences were enriched by a 14 cycle PCR and purified using Celemag clean-up beads before quality assessment and quantification. The libraries were then normalized and pooled prior to being sequenced on the NextSeq 2000 platform (Cat. #20038897, Illumina Inc., San Diego, CA, USA) using the NextSeq 2000 P2 kit (Cat. #20046811, Illumina Inc.). Raw sequencing data were analysed using the Firalink pipeline[8].

### Data management and curation

RNA sequencing (RNAseq) datasets with a relative standard deviation <0.46 and with a number of lncRNAs detected with more than 10 reads in less than 10% of the total FIMICS lncRNAs were excluded. LncRNA data were merged with age, sex, and smoking status for the feature selection process. The missing values of these clinical data were imputed using the missForest function from the missForest R package[29]. Voom-transformed RNAseq data was used for ML analysis[30].

### Machine learning models

The three European cohorts (PrediCOVID, NAPKON, ISARIC4C) were combined and used as a discovery cohort, on which a machine learning procedure was iterated 100 times (Supplementary Fig. 1), following these steps: (1) random selection of 80% of deceased patients and a balanced set of living patients to construct a training dataset; (2) use of the remaining 20% of deceased patients along with a balanced set of the remaining living patients to form a test dataset; (3) identification of differentially expressed lncRNAs in the training dataset with a false discovery rate (FDR) < 0.00001 using the DESeq2 algorithm[31]; (4) feature selection in R on clinical variables (age, sex, and smoking status) and differentially expressed lncRNAs from the training dataset using the Boruta function from the Boruta package[32] and the vif function

from the rms package (https://CRAN.R-project.org/package=rms) with a cut-off of 5 to avoid multi-collinearity; (5) use of repeated (2x) 5-fold cross-validation to fine-tune various machine learning models, including random forest (RF), k-nearest neighbor (kNN), logistic regression (logit), multilayer perceptron (MLP), XGBoost (XGB) and support vector machine (SVM) model in the training dataset using scikit-learn package in Python; (6) evaluation of the model in the test dataset. Features that appeared more than 70 times during the 100 iterations were retained as the selected features that were used to train and evaluate ML models by repeating steps 1, 2, 5 and 6 within 100 iterations with the same seed. The algorithm yielding the model with the highest AUC with the selected features in the test cohort was retained for use in the validation cohort.

The BQC19 cohort was used as the validation cohort. We repeated steps 1 and 2 described above 100 times to split the validation cohort into training and test datasets. In each iteration, a model was trained with the algorithm selected in the discovery cohort using with the features selected there, and evaluated. We also evaluated the selected model on the original imbalanced datasets from the discovery and validation cohort respectively using repeated (20x) 5-fold cross-validation. To test the model robustness, we compared the selected model to the model after adding the top 4 ranked but not selected lncRNAs. The performance metrics, including the AUC, balanced accuracy (accuracy for balanced dataset), sensitivity, and specificity, were reported for the mean and 95% CIs across 100 iterations or the cross-validation. The sensitivity and specificity were determined using 0.5 as the threshold for the predicted class probability.

**Quantitative PCR (qPCR)**

RNA samples extracted from whole blood samples collected in PAXgene tubes were used to assess the expression levels of LEF1-AS1. 200 ng of each RNA sample were reverse transcribed with the High-capacity cDNA reverse transcription kit (ThermoFisher Scientific, Cat # 4368814). To avoid any batch effect, cDNA samples were then randomized in 3 different batches prior to being assessed by quantitative PCR using the CFX-OPUS-96 Dx qPCR device (Biorad, Temse, Belgium) with IQ SYBR Green Supermix (Biorad). Each sample was quantified in duplicate. The following primer sequences designed with the Beacon Designer software (Premier Biosoft) were used for LEF1-AS1: forward 5′- GTCCATGCTATGACCATCTCCA −3′, reverse 5′- ACACGAGTTAAGGCACATTCA −3′; and for SF3A1 which was used as normalizer: forward 5′- GATTGGCCCCAGCAAGCC-3′, reverse 5′- TGCGGAGACAACTGTAGTACG-3′. Splicing Factor 3a Subunit 1 (SF3A1) was chosen as a housekeeping gene for normalization. Expression levels were calculated by the relative quantification method (ΔΔCt) using the CFX Manager 2.1 software (Bio-Rad).

**Statistical analysis**

Continuous and categorical variables were compared with two-sided unpaired Student's $t$-test and Fisher's exact test, respectively. A Mann−Whitney test was used to compare non-normally distributed datasets, as assessed by the Shapiro−Wilk test. Correlation between qPCR and RNAseq data was evaluated using the Spearman's rank test. Cox proportional hazards regression was used to test the association of lncRNAs with survival using the coxph function from the survival R package (https://cran.r-project.org/web/packages/survival/index.html). For survival time, the start date was the date of admission, and the end date was the date of death or the date of discharge. Association between lncRNAs and survival is reported as hazard ratio (HR), along with a measure of precision (95% confidence interval, CI). The significance level was set at 0.05. Kaplan−Meier curves stratified by lncRNA quartile were generated for survival analyses using the ggsurvplot function from the survminer R package (https://cran.r-project.org/web/packages/survminer/index.html).

**Reporting summary**

Further information on research design is available in the Nature Portfolio Reporting Summary linked to this article.

## Data availability

Due to legal and ethical issues related to General Data Protection Regulation guidelines, the data used in this study is available upon request to the COVIRNA consortium. Please email the corresponding author for more details and information about data access (yvan.devaux@lih.lu).

## Code availability

Code accompanying the paper "Development of a long noncoding RNA-based machine learning model to predict COVID-19 in-hospital mortality" is available here: https://doi.org/10.24433/CO.6166592.v1[33].

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

## Acknowledgements

The authors thank all members of COVIRNA project for their contribution: Claude Pelletier, Petr Nazarov, Adriana Voicu, Irina Carpusca, Eric Schordan, Rodwell Mkhwananzi, Stephanie Boutillier, Louis Chauviere, Joanna Michel, Florent Tessier, Reinhard Schneider, Irina Belaur, Wei Gu, Enrico Petretto, Michaela Noseda, Verena Zuber, Pranay Shah, Leonardo Bottolo, Leon de Windt, Emma Robinson, George Valiotis, Tina Hadzic, Federica Margheri, Chiara Gonzi, Detlef Kindgen-Milles, Christian Vollmer, Thomas Dimski, Emin Tahirovic. Further information on the COVIRNA project can be found at https://covirna.eu/. We dedicate this paper to Claude Pelletier who passed away during the timeframe of the COVIRNA project. His invaluable contribution to data analysis is highly recognized and acknowledged. We are thankful to all the participants of the Predi-COVID study. We also acknowledge the involvement of the interdisciplinary and inter-institutional study team that contributed to Predi-COVID. The full list of the Predi-COVID team can be found here: https://sites.lih.lu/the-predi-covid-study/about-us/project-team/. We would like to thank University of Edinburgh DataLoch (https://dataloch.org) and NHS Lothian Bioresource for their support and assistance with this study. This work uses data provided by patients and collected by the NHS as part of their care and support #DataSavesLives. We are extremely grateful to the 2,648 frontline NHS clinical and research staff and volunteer medical students, who collected data in challenging circumstances; and the generosity of the participants and their families for their individual contributions in these difficult times. We also acknowledge the support of Jeremy J Farrar and Nahoko Shindo. The study was carried out using the clinical-scientific infrastructure of NAPKON (Nationales Pandemie Kohorten Netz, German National Pandemic Cohort Network) and NUKLEUS (NUM Klinische Epidemiologie- und Studienplattform, NUM Clinical Epidemiology and Study Platform) of the Network University Medicine (NUM). We gratefully thank all NAPKON sites who contributed patient data and/or biosamples for this analysis. The representatives of NAPKON sites contributing at least 5 per mille to this analysis are (alphabetical order): Bielefeld University, Medical School and University Medical Center OWL, Bielefeld (Alsaad K, Hamelmann E, Heidenreich H, Hornberg C, Kulamadayil-Heidenreich NSA, Maasjosthusmann P, Muna A, Ruwe M, Stellbrink C, Tebbe J), Saarland University, Homburg (Keller A, Walter J), Saarland University Hospital, Homburg (Bals R, Herr C, Krawczyk M, Lensch C, Lepper PM, Riemenschneider M, Smola S, Zemlin M), University Hospital Augsburg, Augsburg (Bader S, Engelmann M, Fuchs A, Langer A, Maerkl B, Messmann H, Muzalyova A, Roemmele C), University Hospital Erlangen, Erlangen (Kraska D, Kremer AE, Leppkes M, Mang J, Neurath MF, Prokosch HU, Schmid J, Vetter M, Willam C, Wolf K), University Hospital Frankfurt, Frankfurt am Main (Arendt C, Bellinghausen C, Cremer S, Groh A, Gruenewaldt A, Khodamoradi Y, Klinsing S, Rohde G, Vehreschild M, Vogl T), University Hospital Hamburg-Eppendorf, Hamburg (Addo M, Almahfoud M, Engels ALF, Jarczak D, Kerinn M, Kluge S, Kobbe R, Petereit S, Schlesner C, Zeller T), University Hospital RWTH Aachen, Aachen (Dahl E, Dreher M, Marx N, Mueller-Wieland D, Wipperfuerth J), University Hospital Regensburg, Regensburg (Brochhausen-Delius C, Burkhardt R, Feustel M, Haag O, Hansch S, Hanses F, Malfertheiner M, Niedermair T, Schuster P, Wallner S), University Hospital Technical University Munich, Munich (Barkey W, Erber J, Fricke L, Lieb J, Michler T, Mueller L, Schneider J, Spinner C, Voit F, Winter C), University Hospital Tuebingen, Tuebingen (Bitzer M, Bunk S, Göpel S, Haeberle H, Kienzle K, Mahrhofer H, Malek N, Rosenberger P, Struemper C, Trauner F), University Hospitals of the Ruhr University Bochum, Bochum (Brinkmann F, Brueggemann Y, Gambichler T, Hellwig K, Luecke T, Reinacher-Schick A, Schmidt WE, Schuette C, Steinmann E, Torres Reyes C), University Medical Center Goettingen, Emergency Department, Goettingen (Blaschke S, Hermanns G, Santibanez-Santana M, Zeh S), University Medical Center Goettingen, Central Biobank, Goettingen (Nussbeck SY), University Medical Center Goettingen, Central Laboratory, Goettingen (Hafke A), University Medicine Essen, Essen (Brochhagen L, Dolff S, Elsner C, Krawczyk A, Madel RJ, Otte M, Witzke O), University Medicine Greifswald, Greifswald (Becker K, Doerr M, Lehnert K, Nauck M, Piasta N, Schaefer C, Schaefer E, Schattschneider M, Scheer C, Stahl D), University Medicine Oldenburg, Oldenburg (Arlt A, Griesinger F, Guenther U, Hamprecht A, Juergens K, Kluge A, Meinhardt C, Meinhardt K, Petersmann A, Prenzel R). This research was supported by the ACBB, the Augsburg Central BioBank (www.biobank-augsburg.de), the CCS Biobank at the University Heart and Vascular Center Hamburg (https://www.uke.de/kliniken-institute/kliniken/kardiologie/forschung/), the Central Biobank Erlangen as a core unit of the Friedrich-Alexander-University Erlangen-Nürnberg in cooperation with the Department of Medicine 1, University Hospital Erlangen, the Central Biobank UMG as a core facility of the University Medical Center Goettingen (Germany),

the HOM.BMB (Biobank Internal Medicine V, Saarland University, Homburg), the Institute of Clinical Chemistry and Laboratory Medicine - Integrated Research Biobank, University Medicine Greifswald, the RWTH centralized Biomaterial Bank (RWTH cBMB) of the Medical Faculty of RWTH Aachen University (https://www.cbmb.ukaachen.de/), the West German Biobank Essen (https://www.uni-due.de/med/biobank/), the ZBR, the Central Biobank Regensburg, the iBioTUM, the Central Interdisciplinary Biomaterial Bank as a Core Unit of the TUM School of Medicine and the University Hospital of the Technical University of Munich, the interdisciplinary Biobank and Database Frankfurt (https://ibdf-frankfurt.de/). We gratefully thank all participating NAPKON infrastructures that contributed to this analysis. The representatives of these NAPKON infrastructures are (alphabetical order): Hannover Unified Biobank, Hannover Medical School, Hannover (Bernemann I, Illig T, Kersting M, Klopp N, Kopfnagel V, Muecke S), Institute of Epidemiology, Helmholtz Center Munich, Munich (Anton G, Kuehn-Steven A, Kunze S, Tauchert MK), University Hospital Frankfurt, Frankfurt (Appel K, Geisler R, Hagen M, Scherer M, Schneider J, Sikdar S, Weirauch T, Wolf L), University Hospital Cologne, Cologne (Brechtel M, Broehl I, Fiedler K, Hopff SM, Laugwitz M, Lee C, Mitrov L, Nunes de Miranda S, Sauer G, Schulze N, Seibel K, Stecher M, Wagner P), University Hospital Wuerzburg, Wuerzburg (Günther K, Haug J, Haug F,), University Hospital Wuerzburg and University of Wuerzburg, Wuerzburg (Fiessler C, Heuschmann PU, Miljukov O, Nürnberger C, Reese JP, Schmidbauer L), University of Wuerzburg, Wuerzburg (Jiru-Hillmann S), University Medicine Greifswald, Greifswald (Bahls T, Hoffmann W, Nauck M, Schaefer C, Schattschneider M, Stahl D, Valentin H), University Medicine Goettingen, Goettingen (Chaplinskaya I, Hans S, Krefting D, Pape C, Rainers M, Schoneberg A, Weinert N), Helmholtz Center Munich, Munich (Kraus M), Charite - Universitaetsmedizin Berlin, Berlin (Lorbeer R, Schaller J). We gratefully thank the NAPKON Steering Committee: University Hospital Giessen and Marburg, Giessen (Herold S), University of Wuerzburg, Wuerzburg (Heuschmann P), Charité - Universitaetsmedizin Berlin, Berlin (Heyder R), University Medicine Greifswald, Greifswald (Hoffmann W), Hannover Unified Biobank, Hannover Medical School, Hannover (Illig T), University Hospital Schleswig-Holstein, Kiel (Schreiber S), University Hospital Cologne and University Hospital Frankfurt, Cologne and Frankfurt (Vehreschild JJ), Charité - Universitaetsmedizin Berlin, Berlin (Witzenrath M). We greatly thank all ISARIC4C investigators. The full ISARIC4C Investigator list can be found at this link.This work uses Data / Material provided by patients and collected by the NHS as part of their care and support #DataSavesLives. The Data / Material used for this research were obtained from ISARIC4C. The COVID-19 Clinical Information Network (CO-CIN) data was collated by ISARIC4C Investigators. This work was made possible through open sharing of data and samples from the Biobanque québécoise de la COVID-19 (BQC19). We thank all participants of the BQC19 for their contribution. This work was supported by the EU Horizon 2020 project COVIRNA (grant agreement # 101016072) The Predi-COVID study was supported by the Luxembourg National Research Fund (FNR) (Predi-COVID, grant number 14716273), the André Losch Foundation and by European Regional Development Fund (FEDER, convention 2018-04-026-21). The project National Pandemic Cohort Network (NAPKON) is part of the Network University Medicine (NUM), funded by the German Federal Ministry of Education and Research (BMBF) (FKZ: 01KX2121). Parts of the infrastructure of the Würzburg study site were supported by the Bavarian Ministry of Research and Art to support Corona research projects. Parts of the NAPKON project suite and study protocols of the Cross-Sectoral Platform are based on projects funded by the German Center for Infection Research (DZIF). Data and Material provision for ISARIC4C was supported by grants from: the National Institute for Health Research (NIHR; award CO-CIN-01), the Medical Research Council (MRC; grant MC_PC_19059), and by the NIHR Health Protection Research Unit (HPRU) in Emerging and Zoonotic Infections at University of Liverpool in partnership with Public Health England (PHE), (award 200907), NIHR HPRU in Respiratory Infections at Imperial College London with PHE (award 200927), Liverpool Experimental Cancer Medicine Center (grant C18616/A25153), NIHR Biomedical Research Center at Imperial College London (award IS-BRC-1215-20013), and NIHR Clinical Research Network providing infrastructure support. The Biobanque québécoise de la COVID-19 was funded by the Fonds de recherche du Québec - Santé, Génome Québec, the Public Health Agency of Canada and, as of March 2022, the Ministère de la Santé et des Services Sociaux. Project no. RRF-2.3.1-21-2022-00003 has been implemented with the support provided by the European Union. The 2020-1.1.5-GYORSÍTÓSÁV-2021-00011 project was funded by the Ministry for Innovation and Technology with support from the National Research Development and Innovation Fund under the 2020-1.1.5-GYORSÍTÓSÁV call program. The project was supported by grants from the National Research, Development and Innovation Office (NKFIH) of Hungary, 2020-1.1.6-JÖVÖ–2021-00013. Y.D. is funded by the EU Horizon 2020 project COVIRNA (grant agreement # 101016072), the National Research Fund (grants # C14/BM/8225223, C17/BM/11613033 and COVID-19/2020-1/14719577/miRCOVID), the Ministry of Higher Education and Research, and the Heart Foundation-Daniel Wagner of Luxembourg. F.M. is supported by the Italian Ministry of Health projects "Ricerca Corrente 2023", and POS T4 CAL.HUB.RIA, cod. T4-AN-09.

## Author contributions

Y.D., L.Z. and A.I.L. led the writing and editing. Y.D., L.Z., K.H. analyzed the data. A.B., V.M., S.R., J.J.V., B.L.D., M.D., G.F., M.O. provided human samples. All others including M.Sh, V.S., M.A., P.K.S., C.E., F.M., S.G., L.B., T.P., M.L., M.Sc, M.R., M.J., T.B., B.A., P.F., B.B., O.W., G.S., S.K., R.W., N.L.M., H.F. provided background information, intellectual contributions, editing, and/or writing of the manuscript.

## Competing interests

Y.D. holds patents and licensing agreements related to the use of RNAs for diagnostic and therapeutic purposes, and is SAB member of Firalis SA. P.F. is the founder and CEO of Pharmahungary Group, a group of R&D companies. J.J.V. has personal fees from Merck / MSD, Gilead, Pfizer, Astellas Pharma, Basilea, German Center for Infection Research (DZIF), University Hospital Freiburg/ Congress and Communication, Academy for Infectious Medicine, University Manchester, German Society for Infectious Diseases (DGI), Ärztekammer Nordrhein, University Hospital Aachen, Back Bay Strategies, German Society for Internal Medicine (DGIM), Shionogi, Molecular Health, Netzwerk Universitätsmedizin, Janssen, NordForsk, Biontech, APOGEPHA. J.J.V. has grants from Merck / MSD, Gilead, Pfizer, Astellas Pharma, Basilea, German Center for Infection Research (DZIF), German Federal Ministry of Education and Research (BMBF), Deutsches Zetrum für Luft- und Raumfahrt (DLR), University of Bristol, Rigshospitalet Copenhagen, Network University Medicine. L.B. declares to have acted as a SAB member of Sanofi, Ionnis, MSD and NovoNordisk; to have received speaker fees from Sanofi, Bayer and AB-Biotics SA and to have founded the spin-off Ivastatin Therapeutics S.L. (all unrelated to this work). T.P. declares to have received speaker fees from AB-Biotics SA, and to be a co-founder of the Spin-off Ivastatin Therapeutics S.L. (all unrelated to this work). M.S. received funding from Pfizer Inc. and from Owkin for projects not related to this research. N.L.M. has received honoraria from Abbott Diagnostics, Siemens Healthineers, Roche Diagnostics and LumiraDx. The University of Edinburgh has received research grants from Abbott Diagnostics, Novo Nordisk, and Siemens Healthineers. H.F. is the founder and owner of Firalis SA, a company commercializing the FIMICS panel. He holds patents and licenses for the use of RNAs as biomarkers and therapeutic targets. The remaining authors declare no competing interests

## Additional information

[1]Cardiovascular Research Unit, Department of Precision Health, Luxembourg Institute of Health, Strassen, Luxembourg. [2]Bioinformatics Platform, Luxembourg Institute of Health, Strassen, Luxembourg. [3]Faculty of Engineering and Natural Sciences, International University of Sarajevo, Sarajevo, Bosnia and Herzegovina. [4]Department of Human Genetics, McGill University, Montréal, QC, Canada. [5]The Meakins-Christie Laboratories at the Research Institute of the McGill University Heath Centre Research Institute, & Department of Medicine, Faculty of Medicine, McGill University, Montréal, QC, Canada. [6]Luxembourg Center for Systems Biomedicine, University of Luxembourg, Belval, Luxembourg. [7]National Heart and Lung Institute, Imperial College London, London, England, UK. [8]Molecular Cardiology Laboratory, IRCCS Policlinico San Donato, Milan, Italy. [9]Cardiovascular Program-ICCC, Institut d'Investigació Biomèdica Sant Pau (IIB SANT PAU); CIBERCV, Autonomous University of Barcelona, Barcelona, Spain. [10]Department of Intelligent Systems, Jozef Stefan Institute, Ljubljana, Slovenia. [11]Group Genetical Statistics and Biomathematical Modelling, Institute for Medical Informatics, Statistics and Epidemiology, University of Leipzig, Leipzig, Germany. [12]Medical University of Dusseldorf, Dusseldorf, Germany. [13]HUN-REN–SU System Pharmacology Research Group, Department of Pharmacology and Pharmacotherapy, Semmelweis University, Budapest, Hungary; Pharmahungary Group, Szeged, Hungary. [14]Medical Department 2 (Hematology/Oncology and Infectious Diseases), Center for Internal Medicine, Goethe University Frankfurt, University Hospital, Frankfurt, Germany. [15]University of Cologne, Faculty of Medicine and University Hospital Cologne, Cologne, Germany. [16]Department I of Internal Medicine, Center for Integrated Oncology Aachen Bonn Cologne Duesseldorf, Cologne, Germany. [17]German Centre for Infection Research (DZIF), partner site Bonn-Cologne, Cologne, Germany. [18]Institute of Epidemiology, Helmholtz Center Munich, Munich, Germany. [19]Department of Internal Medicine B, University Medicine Greifswald, Greifswald, Germany; German Centre of Cardiovascular Research (DZHK), Greifswald, Germany. [20]Department of Infectious Diseases, West German Centre of Infectious Diseases, University Hospital Essen, University of Duisburg-Essen, Essen, Germany. [21]Firalis SA, Huningue, France. [22]Centre for Cardiovascular Science, The Queen's Medical Research Institute, University of Edinburgh, Edinburgh, Scotland. [23]CARIM Institute and Department of Pathology, University of Maastricht, Maastricht, The Netherlands. [24]Deep Digital Phenotyping Research Unit, Department of Precision Health, Luxembourg Institute of Health, Strassen, Luxembourg. [25]Department of Infection and Immunity, Luxembourg Institute of Health, Esch-Sur-Alzette, Luxembourg. [26]Department of Dermatology and Allergy Center, Odense Research Center for Anaphylaxis (ORCA), University of Southern Denmark, Odense, Denmark. [27]Centre for Cardiovascular Science, University of Edinburgh, Edinburgh, UK. [28]Usher Institute, University of Edinburgh, Edinburgh, UK. ✉e-mail: yvan.devaux@lih.lu

