## [Peer Review File · Nature Communications]

Development of a long noncoding RNA-based machine learning model to predict COVID-19 in-hospital mortalityREVIEWER COMMENTS

Reviewer #1 (Remarks to the Author):

This is an interesting study and with exciting results of translational value. However, some details are missing from assessing the value of conclusions.

The demographics are missing ethnic data which research has shown are drastically affect the risk assessing algorithms ([https://www.thelancet.com/journals/lanepi/article/PIIS2666-7762\(22\)00105-3/fulltext](https://www.thelancet.com/journals/lanepi/article/PIIS2666-7762(22)00105-3/fulltext)). Key data such as prior medical history, or if these patients were receiving ongoing treatment for other conditions is important to address or at least mention in limitations if that could bias some results.

Age, smoking, and sex with differential gene expression data seems to be the input for feature selection. However, missing data was imputed without specifying its percentage. For example, was 40% age variable missing in cohort A. These characteristics need to be mentioned for assessment of the lncRNA as a potential biomarker.

Was the data input data imputed for the validation cohort, if yes, then what were missing proportions. These parameters could skew the results in favor of the biomarker.

Is LEF1-AS1 correlated to age? or cancer diagnosis? what was the nature of COVID-19 symptoms, mild or extreme symptomology such as needing intensive care or oxygen therapy etc?

Why was survival group limited to ISARIC4C subgroup?

How much do the AUC and other metrics change when missing variables are not imputed?

Reviewer #1 (Remarks on code availability):

I did not find the file with code.

Reviewer #2 (Remarks to the Author):

Thank you for the opportunity to review this interesting manuscript exploring the ability of age combined with a novel long non-coding RNA (LEF1-AS1) as a risk prediction model for mortality in patients admitted to hospital with SARS-CoV-2 infection. This risk stratification model was developed and validated across 1286 patients in 13 countries. The potential to measure RNA LEF1-AS1 using qualitative PCR is described.

Combining age and RNA LEF1-AS1 appears to have good to excellent predictive performance in the validation cohort and I have no major concerns regarding the statistical methodology described. However, the performance in the validation cohort must be interpreted with caution of the development and validation cohorts are similar demographically - unfortunately the basic demographics of each cohort was was not included. There are also no details regarding the period over which samples were taken, or the proportion of patients who were vaccinated.

The authors also do not include reference to the TRIPOD guidelines, which are the gold-standard for reporting risk stratification models.

In terms of performance, a number of high-quality risk stratification models for predicting in-patient mortality for patients admitted with SARS-CoV-2 infection already exist (see Wynants et al living review published in the BMJ). What is their comparative performance compared to the novel model described? Does age and RNA LEF1-AS1 outperform these models, enhancing impact and highlighting its importance clinically? How do they compare not only in AUC value (which says little about performance across the range of risk and interpretation in isolation should be avoided) but in assessing calibration and possible 'real-world' performance through DCA analysis?

Reviewer #3 (Remarks to the Author):

This paper nicely shows that increased expression of lncRNA LEF1-AS1 is associated with a lower risk of in-hospital mortality in a large cohort of COVID-19 patients, and that LEF1-AS1 expression, taken together with age, is sufficient to train a reasonable ML model for predicting such mortality. While it was already well established that age is a risk factor for severe COVID outcomes, and the authors had already published some preliminary evidence from a much smaller population suggesting that LEF1-AS1 expression is associated with poor COVID-19 outcomes, this study is important for several reasons. Firstly, the scale of the cohorts under study is significantly larger, so the conclusions are more likely to be robust. Secondly, the role of lncRNAs in human biology is still being unraveled, and this study provides some important clues about their role in the immune response. Finally, the classifier performs relatively well, suggesting possible clinical utility in the future. I think this study should be accepted, but I would like to see a few small enhancements:

I would like to better understand the extent to which LEF1-AS1 expression is independent of age and gender, which are both known risk factors for more severe COVID-19 outcomes. How correlated is LEF1-AS1 expression with these demographic variables? Some studies in other disease areas have found that it is not correlated with age, but it would be nice to see this for the cohorts used in this study.

<https://www.ncbi.nlm.nih.gov/pmc/articles/PMC6433713/>

<https://www.ncbi.nlm.nih.gov/pmc/articles/PMC6765338/>

Another way to further tease apart the relative importance of lncRNA expression in the model predictions would be to show a Shapley beeswarm plot of the feature weights for the models. Ideally we could see this for one of the larger models with additional lncRNAs.

<https://shap.readthedocs.io>

It would be nice to see if this model could be improved with other features, such as BMI and gender.

It might be possible to improve the model performance (by a modest amount) using a gradient-boosted tree model. Since this is a very simple code change, it would probably be worthwhile. It would be nice to see if using xgboost might allow the model to glean additional information from some of the other lncRNAs in the panel.

<https://scikit-learn.org/stable/modules/generated/sklearn.ensemble.GradientBoostingClassifier.html>

<https://xgboost.readthedocs.io/en/stable/tutorials/model.html>

Reviewer #3 (Remarks on code availability):

I may have missed something, but I didn't see a link to the code.

Reviewer #1

This is an interesting study and with exciting results of translational value. However, some details are missing from assessing the value of conclusions. The demographics are missing ethnic data which research has shown are drastically affect the risk assessing algorithms ([https://www.thelancet.com/journals/lanepi/article/PIIS2666-7762\(22\)00105-3/fulltext](https://www.thelancet.com/journals/lanepi/article/PIIS2666-7762(22)00105-3/fulltext)). Key data such as prior medical history, or if these patients were receiving ongoing treatment for other conditions is important to address or at least mention in limitations if that could bias some results.

Authors' reply: the authors thank this reviewer for his/her careful evaluation of their manuscript and his/her comments that helped improving its quality.

As requested, we have added in Table 1 the demographic and clinical characteristics which were available at baseline in all four patient cohorts included in the study: diabetes, chronic lung disease, cardiovascular disease, oxygen therapy. Furthermore, we have included characteristics of patients specific to the different cohorts in Supplementary Tables 1 and 2. Being a retrospective study of cohorts built in an emergency situation during the first wave of the COVID-19 pandemic, limited and sometimes heterogeneous data were available from the different patient cohorts. Prior medical history and ongoing treatments for other conditions were unavailable. Ethnicity was available in the NAPKON cohort only, in which most of patients were Caucasian, and no apparent association between ethnicity and survival was found. In this cohort, vaccinated people had a lower risk of death as compared to non-vaccinated people. Vaccination data was unavailable in other cohorts. The manuscript has been amended as follows to reflect these points, page 8 lines 24-37: *"In all cohorts, patients who died in hospital were older than survivors, more often had cardiovascular disease, and more often received oxygen therapy. Being a male was associated with a higher risk of death in the merged European cohorts. Diabetes and chronic lung disease were also risk factors in this cohort. Patients in the Canadian cohort were older, were more often females and were less often smokers than patients in the merged European cohorts. Supplementary Table 1 shows the characteristics of the three European cohorts individually, together with the nature of common COVID-19 symptoms across cohorts. The PrediCOVID cohort had younger patients than the two other cohorts and none of them died during the follow-up period. There were more smokers at the time of enrolment in the PrediCOVID cohort than in the NAPKON and ISARIC4C cohorts. Common baseline symptoms across the European cohorts included fever, headache, cough and dyspnea, which were less frequent in survivors (Supplementary Table 1). Ethnicity data was available in the NAPKON cohort, in which most patients were Caucasian and no apparent association between ethnicity and survival was found (Supplementary Table 1). In this cohort, vaccinated people had a lower risk of death as compared to non-vaccinated people (Supplementary Table 1). Vaccination data was unavailable in other cohorts."*

Following the addition of new demographic and clinical data, we have tested how incorporating this new information (diabetes, chronic lung disease, cardiovascular disease, oxygen therapy, which were available in all patients) influences the result of the present study. Incorporation of these variables to the feature selection step did not change the initial results, that age and LAF1-AS1 were selected as the best predictors of survival. Oxygen therapy was selected as the third best predictor, yet this variable did not significantly improve the performance of prediction model in the balanced (AUC=0.84[0.82-0.86], p=0.11 for comparison with the model without oxygen therapy) and imbalanced (AUC=0.83[0.82-0.84], p=0.91 for comparison with the model without oxygen therapy) discovery dataset. Figure 2A has been amended accordingly and the Results section of the manuscript page 9 lines 21-24 has been modified as follows: *"Adding the third best predictor selected during the feature selection step did not improve the performance of the prediction model in the balanced (AUC 0.84 [0.82 – 0.86]. p = 0.11 for comparison with the model without oxygen therapy) and imbalanced (AUC = 0.83 [0.82-0.84], p = 0.91 for comparison with the model without oxygen therapy) discovery dataset"*.

Age, smoking, and sex with differential gene expression data seems to be the input for feature selection. However, missing data was imputed without specifying its percentage. For example, was 40% age variable missing in cohort A. These characteristics need to be mentioned for assessment of the lncRNA as a potential biomarker.

Authors' reply: age, smoking, and sex with gene expression data indeed constituted the initial input for feature selection. As stated in the previous comment, addition of new demographic and clinical data in the feature selection process did not influence the result of the present study, that age and LAF1-AS1 were selected as the best predictors of survival. As far as missing data are concerned, in the discovery cohort, all patients had sex information. Four (0.5%), 150 (18.7%), 16 (2.0%), 15 (18.7%), 19 (2.4%) and 12 (1.5%) patients had missing values on age, smoking, diabetes, chronic lung disease, cardiovascular disease and oxygen therapy, respectively. This information has been included in Table 1. The following sentence has been added to the Results section page 8 line 22: *"Missing data are indicated and were imputed using missForest"*. There were no missing gene expression data.

Was the data input data imputed for the validation cohort, if yes, then what were missing proportions. These parameters could skew the results in favor of the biomarker.

Authors' reply: age was the only clinical variable selected for machine learning from the validation cohort. Since in this cohort, all patients had age information, there was no imputation of missing data and our results are not skewed in favour of the biomarker. The numbers of patients with other missing data (smoking, diabetes ...) are indicated in Table 1 for this cohort.

Is LEF1-AS1 correlated to age? or cancer diagnosis? What was the nature of COVID-19 symptoms, mild or extreme symptomology such as needing intensive care or oxygen therapy etc?

Authors' reply: LEF1-AS1 was weakly and negatively correlated to age in both discovery ($r=-0.21$) and validation cohorts ($r=-0.35$). This info has been added in the Results section page 9 lines 9-11 as follows: *"There was a significant albeit moderate negative correlation between age and LEF1-AS1 in this cohort (Figure 2D), as well as in the Validation cohort ($r=-0.35$, $p < 0.01$)"*. A panel D to Figure 2 has been added showing correlation between age and LEF1-AS1 in the Discovery cohort.

We had information on cancer diagnosis in the NAPKON and BQC19 cohorts, which has been added to Supplementary Table 1 for NAPKON cohort. The expression of LEF1-AS1 was associated with cancer diagnosis with an odds ratio of 0.71[0.55-0.90] and 0.66 [0.52-0.84] in the NAPKON and BQC19 cohorts, respectively. This sentence has been added in the Results section page 9 lines 13-14.

Common baseline COVID-19 symptoms across each European cohort have been added to Supplementary Table 1. We could not include these symptoms for the merged European cohorts in Table 1 since the 3 EU cohorts used different symptomatology assessments. The following sentence has been added in the Results section page 8 lines 32-34 to reflect this additional information: *"Common baseline symptoms across the European cohorts included fever, headache, cough and dyspnea, which were less frequent in survivors (Supplementary Table 1)"*.

Why was survival group limited to ISARIC4C subgroup?

Authors' reply: all patient subgroups were included in the survival analysis using ML, in which patients were classified as survivors or non-survivors. Since we had data on time to death only for the ISARIC4C and BQC19 subgroups, the survival analysis using Cox regression and Kaplan-Meier curves could be conducted only in

these two subgroups (Figures 4 and 5). The limitation section of the manuscript has been amended to reflect this point as follows, page 13 lines 6-8: *“Fourth, survival analysis using Cox regression and Kaplan-Meier curves could be conducted only in the ISARIC4C and BQC19 subgroups for which we had data on time to death.”*

How much do the AUC and other metrics change when missing variables are not imputed?

Authors’ reply: additional feature selection and ML analyses have been conducted without prior imputation and data have been gathered in a new Supplementary Table 3. Results show that imputing missing variables for age (which concerns only 4 patients of the Discovery cohort, see Table 1) did not affect the predictive performance of the model with age and LEF-AS1. As stated before, there were no missing data for LEF1-AS1. The following sentence has been added in the Results section page 9 lines 39-41: *“Missing data imputation did not significantly influence results since the MLP model run without prior imputation of missing age data (concerning only 4 patients of the Discovery cohort, Table 1) reached an AUC of 0.83 and a balanced accuracy of 0.78 (Supplementary Table 3)”*.

I did not find the file with code.

Authors’ reply: The ML code is now provided as a supplementary file to the submission and has been uploaded in the Code Ocean platform (<https://codeocean.com/capsule/4420803/tree/v1>). Data availability and Code availability sections have been added to the manuscript page 25 lines 32-34 and page 26 lines 1-3.

Reviewer #2

Thank you for the opportunity to review this interesting manuscript exploring the ability of age combined with a novel long non-coding RNA (LEF1-AS1) as a risk prediction model for mortality in patients admitted to hospital with SARS-CoV-2 infection. This risk stratification model was developed and validated across 1286 patients in 13 countries. The potential to measure RNA LEF1-AS1 using qualitative PCR is described.

Authors' reply: the authors thank this reviewer for his/her careful evaluation of their manuscript and his/her comments that helped improving its quality.

Combining age and RNA LEF1-AS1 appears to have good to excellent predictive performance in the validation cohort and I have no major concerns regarding the statistical methodology described. However, the performance in the validation cohort must be interpreted with caution of the development and validation cohorts are similar demographically - unfortunately the basic demographics of each cohort was not included. There are also no details regarding the period over which samples were taken, or the proportion of patients who were vaccinated.

Authors' reply: As requested, we have added in Table 1 the demographic and clinical characteristics which were available at baseline in all four patient cohorts included in the study: diabetes, chronic lung disease, cardiovascular disease, oxygen therapy. Furthermore, we have included characteristics of patients specific to the different cohorts in Supplementary Tables 1 and 2. Being a retrospective study of cohorts built in an emergency situation during the first wave of the COVID-19 pandemic, limited and sometimes heterogeneous data were available from the different patient cohorts. Prior medical history and ongoing treatments for other conditions were unavailable. Ethnicity was available in the NAPKON cohort only, in which most of patients were Caucasian, and no apparent association between ethnicity and survival was found. In this cohort, vaccinated peoples had a lower risk of death as compared to non-vaccinated peoples. Vaccination data was unavailable in other cohorts. The manuscript has been amended as follows to reflect these points, page 8 lines 24-37: *"In all cohorts, patients who died in hospital were older than survivors, more often had cardiovascular disease, and more often received oxygen therapy. Being a male was associated with a higher risk of death in the merged European cohorts. Diabetes and chronic lung disease were also risk factors in this cohort. Patients in the Canadian cohort were older, were more often females and were less often smokers than patients in the merged European cohorts. Supplementary Table 1 shows the characteristics of the three European cohorts individually, together with the nature of common COVID-19 symptoms across cohorts. The PrediCOVID cohort had younger patients than the two other cohorts and none of them died during the follow-up period. There were more smokers at the time of enrolment in the PrediCOVID cohort than in the NAPKON and ISARIC4C cohorts. Common baseline symptoms across the European cohorts included fever, headache, cough and dyspnea, which were less frequent in survivors (Supplementary Table 1). Ethnicity data was available in the NAPKON cohort, in which most patients were Caucasian and no apparent association between ethnicity and survival was found (Supplementary Table 1). In this cohort, vaccinated people had a lower risk of death as compared to non-vaccinated people (Supplementary Table 1). Vaccination data was unavailable in other cohorts."*

Following the addition of new demographic and clinical data, we have tested how incorporating this new information (diabetes, chronic lung disease, cardiovascular disease, oxygen therapy, which were available in all patients) influences the result of the present study. Incorporation of these variables to the feature selection step did not change the initial results, that age and LAF1-AS1 were selected as the best predictors of survival. Oxygen therapy was selected as the third best predictor, yet this variable did not significantly improve the performance of prediction model in the balanced (AUC=0.84[0.82-0.86], p=0.11 for comparison with the model without oxygen therapy) and imbalanced (AUC=0.83[0.82-0.84], p=0.91 for comparison with the model without oxygen therapy) discovery dataset. Figure 2A has been amended accordingly and the

Results section of the manuscript page 9 lines 21-24 has been modified as follows: *“Adding the third best predictor selected during the feature selection step did not improve the performance of the prediction model in the balanced (AUC 0.84 [0.82 – 0.86], $p = 0.11$ for comparison with the model without oxygen therapy) and imbalanced (AUC = 0.83 [0.82-0.84], $p = 0.91$ for comparison with the model without oxygen therapy) discovery dataset”.*

Periods over which samples were collected have been added in the Methods section page 5 lines 15-17 as follows: *“Periods of patient enrolment and biological samples collection were as follows: May 2020 - Present for PrediCOVID, July 2020 - Present for NAPKON, February 2020 - September 2020 for ISARIC4C, March 2020 - Present for BQC19.”*

Many patients included in the present study were enrolled in the first phase of the pandemic during which there was limited/no vaccine available. Vaccination status of study participants was available for the NAPKON cohort only and has been added to Supplementary Table 1. The Results section page 8 lines 36-37 has been amended as follows: *“In this cohort, vaccinated peoples had a lower risk of death as compared to non-vaccinated peoples (Supplementary Table 1). Vaccination data was unavailable in other cohorts.”* In addition, the following sentence has been added in the Limitations chapter of the Discussion section page 12, Line 38-40: *“Also, there was limited information on vaccination status due to the fact that there were no widely available vaccines at the time of study enrolment.”*

The authors also do not include reference to the TRIPOD guidelines, which are the gold-standard for reporting risk stratification models.

Authors’ reply: The TRIPOD guideline document has been filled in and attached with the submission.

In terms of performance, a number of high-quality risk stratification models for predicting in-patient mortality for patients admitted with SARS-CoV-2 infection already exist (see Wynants et al living review published in the BMJ). What is their comparative performance compared to the novel model described? Does age and RNA LEP1-AS1 outperform these models, enhancing impact and highlighting its importance clinically? How do they compare not only in AUC value (which says little about performance across the range of risk and interpretation in isolation should be avoided) but in assessing calibration and possible 'real-world' performance through DCA analysis?

Authors’ reply: We thank this reviewer for this important comment. As reviewed and appraised by Wynants and colleagues in their live review (BMJ 2020; Apr 7), most of the diagnostic and prognostic models published so far for COVID-19 have a high risk of bias. The authors identified eight models predicting mortality. The performance of a previously published model (which included age, sex, CRP and LDH levels) alongside our MLP model, has been analysed using 2 of our study cohorts (NAPKON and BQC19). The results of this comparison are presented in Supplementary Figure 6. From this figure we can observe that when applied to BQC19 data, both our model and the other model perform similarly (AUC 0.83 vs 0.85, respectively). When applied to NAPKON data, our model outperformed the other model (AUC 0.82 vs 0.78, respectively). As the data included in our model (age and SEQ0235 expression) are continuous, we used the Brier score instead of DCA analysis to assess calibration and possible real-world performance of our model. The Brier score is the mean squared difference between the predicted probability and the actual outcome. A lower Brier score illustrates a better-calibrated model. The Brier score for both models in BQC19 data were comparable, however our model was better calibrated than the other model using NAPKON data (Supplementary Figure 6). Furthermore, we have added the Brier score in Tables 2, 3 and 4 as an indication of models calibration.

Overall, the main advantages of our study rely on its methodological aspects which reduce the risks of bias. We conducted a multi-center and well powered study, with patient numbers well above previous studies.

We have used a machine learning pipeline including feature selection and testing of multiple machine learning models with Discovery and Validation cohorts. In each cohort, we have evaluated models on the imbalanced datasets using twenty times repeated 5-fold cross validation.

Because the study end-point was survival, we have indeed completed AUC values with survival analyses with COX regression and Kaplan Meier curves.

These points have been addressed by amendments

- of the Results section page 10 lines 1-8: *“We compared the predictive performance of the MLP model with age and LEF1-AS1 to a previously published four-parameter model involving age, sex, C-reactive protein (CRP) and lactate dehydrogenase (LDH). As shown in Supplementary Figure 6, our MLP model and the four-parameter model had similar capacity to predict mortality in the BQC19 cohort (AUC 0.83 vs 0.85, respectively). Our MLP model outperformed the four-parameter model in the NAPKON cohort (AUC 0.82 vs 0.78, respectively). Brier score analysis was used to assess the calibration of our MLP model, where a lower Brier score indicates a more calibrated model. This analysis revealed a similar (for BQC-19 data) and a lower score (for NAPKON data) for our MLP model compared to the previously published four-parameter model (Supplementary Figure 6).”*
- of the Discussion section page 11 lines 20-29: *“Models to predict mortality of COVID-19 patients have been previously developed, yet they suffer from a high risk of bias²⁰. The MLP model reported in the present study with only two features (age and LEF1-AS1) showed similar predictive performance in the BQC-19 cohort and higher performance in the NAPKON cohort compared to a model including age, sex, CRP and LDH. As compared to previous reports²⁰, the strength of our study relies on its methodological aspects which reduce the risks of bias. We conducted a multi-center and well powered study, with patient numbers well above previous studies. We have used a machine learning pipeline including feature selection and testing of multiple machine learning models with Discovery and Validation cohorts, each split into training and testing subgroups. In each cohort, we have evaluated models on the imbalanced datasets using twenty times repeated 5-fold cross validation.”*

Reviewer #3

This paper nicely shows that increased expression of lncRNA LEF1-AS1 is associated with a lower risk of in-hospital mortality in a large cohort of COVID-19 patients, and that LEF1-AS1 expression, taken together with age, is sufficient to train a reasonable ML model for predicting such mortality. While it was already well established that age is a risk factor for severe COVID outcomes, and the authors had already published some preliminary evidence from a much smaller population suggesting that LEF1-AS1 expression is associated with poor COVID-19 outcomes, this study is important for several reasons. Firstly, the scale of the cohorts under study is significantly larger, so the conclusions are more likely to be robust. Secondly, the role of lncRNAs in human biology is still being unraveled, and this study provides some important clues about their role in the immune response. Finally, the classifier performs relatively well, suggesting possible clinical utility in the future. I think this study should be accepted, but I would like to see a few small enhancements:

Authors' reply: the authors thank this reviewer for his/her careful evaluation of their manuscript and his/her comments that helped improving its quality.

I would like to better understand the extent to which LEF1-AS1 expression is independent of age and gender, which are both known risk factors for more severe COVID-19 outcomes. How correlated is LEF1-AS1 expression with these demographic variables? Some studies in other disease areas have found that it is not correlated with age, but it would be nice to see this for the cohorts used in this study.

<https://www.ncbi.nlm.nih.gov/pmc/articles/PMC6433713/>

<https://www.ncbi.nlm.nih.gov/pmc/articles/PMC6765338/>

Authors' reply: of age and sex, only age was retained in the final predictive model with LEF1-AS1. We assessed the correlation between age and LEF1-AS1 in both Discovery and Validation cohorts. A significant correlation was found in both. As far as sex is concerned, there was a significant difference in LEF1-AS1 between males and females. This data has been added as panels D-E in Figure 2 and in the Results section page 9 lines 9-13 as follows: *"There was a significant albeit moderate negative correlation between age and LEF1-AS1 in this cohort (Figure 2D), as well as in the Validation cohort ($r=-0.35$, $p < 0.01$). Also, LEF1-AS1 was differentially expressed between males and females in the Discovery (Figure 2E) and in the Validation cohort ($p < 0.01$ and $p = 0.02$, respectively)."*

Another way to further tease apart the relative importance of lncRNA expression in the model predictions would be to show a Shapley beeswarm plot of the feature weights for the models. Ideally we could see this for one of the larger models with additional lncRNAs.

<https://shap.readthedocs.io>

Authors' reply: as suggested, we have included Shapley beeswarm plots as Supplementary Figure 2 showing that in both Discovery and Validation cohorts higher age and lower expression of SEQ0235 led to positive SHAP values and thus had positive impacts on model output. The following sentence has been included in the Results section page 9 lines 14-16: *"The Shapley beeswarm plots shown in Supplementary Figure 2 attest that higher age and lower expression of LEF1-AS1 led to positive SHAP values and thus had positive impacts on model output."*

It would be nice to see if this model could be improved with other features, such as BMI and gender.

Authors' reply: the model with age and LEF1-AS provided the optimal predictive value. Addition of sex did not further improve the model. BMI data was unavailable in the NAPKON and BQC19 cohorts. This info is now included in Supplementary Figure 5B and in the Results section page 9 lines 36-39 as follows: *"Adding combinations of the top 4 non selected lncRNAs, or sex and/or the other 2 features which were selected more*

than 40 times in the final model (oxygen therapy and SEQ0986, Figure 2A) did not significantly improve the model performance (Supplementary Figure 5)."

It might be possible to improve the model performance (by a modest amount) using a gradient-boosted tree model. Since this is a very simple code change, it would probably be worthwhile. It would be nice to see if using xgboost might allow the model to glean additional information from some of the other lncRNAs in the panel.

**<https://scikit-learn.org/stable/modules/generated/sklearn.ensemble.GradientBoostingClassifier.html>
<https://xgboost.readthedocs.io/en/stable/tutorials/model.html>**

Authors' reply: as suggested, we have tested the XGB xgboost model (now included in Machine Learning models Section, Page 6, line 23) and no improvement of performance was observed. This data has been added to Tables 2 and 3 and in the Results section page 8 line 39-40 as follows: "RF, kNN, Logit, MLP, SVM, XGB"

I may have missed something, but I didn't see a link to the code.

Authors' reply: The ML code is now provided as a supplementary file to the submission and has been uploaded in the Code Ocean platform (<https://codeocean.com/capsule/4420803/tree/v1>). Data availability and Code availability sections have been added to the manuscript page 25 lines 32-34 and page 26 lines 1-3.

REVIEWERS' COMMENTS

Reviewer #1 (Remarks to the Author):

Authors have adequately addressed my comments.

Reviewer #2 (Remarks to the Author):

All responses to reviewers are well written and fully explained. No further comments.

Reviewer #3 (Remarks to the Author):

I am satisfied with the authors' revisions and am happy for the paper to be published.